# Role of eIF5A in Mitochondrial Function

**DOI:** 10.3390/ijms23031284

**Published:** 2022-01-24

**Authors:** Marina Barba-Aliaga, Paula Alepuz

**Affiliations:** 1Instituto de Biotecnología y Biomedicina (Biotecmed), Universitat de València, 46100 València, Spain; 2Departamento de Bioquímica y Biología Molecular, Facultad de Ciencias Biológicas, Universitat de València, 46100 València, Spain

**Keywords:** eIF5A, mitochondria, translation, spermidine, mitochondrial respiration, OXPHOS, TCA

## Abstract

The eukaryotic translation initiation factor 5A (eIF5A) is an evolutionarily conserved protein that binds ribosomes to facilitate the translation of peptide motifs with consecutive prolines or combinations of prolines with glycine and charged amino acids. It has also been linked to other molecular functions and cellular processes, such as nuclear mRNA export and mRNA decay, proliferation, differentiation, autophagy, and apoptosis. The growing interest in eIF5A relates to its association with the pathogenesis of several diseases, including cancer, viral infection, and diabetes. It has also been proposed as an anti-aging factor: its levels decay in aged cells, whereas increasing levels of active eIF5A result in the rejuvenation of the immune and vascular systems and improved brain cognition. Recent data have linked the role of eIF5A in some pathologies with its function in maintaining healthy mitochondria. The eukaryotic translation initiation factor 5A is upregulated under respiratory metabolism and its deficiency reduces oxygen consumption, ATP production, and the levels of several mitochondrial metabolic enzymes, as well as altering mitochondria dynamics. However, although all the accumulated data strongly link eIF5A to mitochondrial function, the precise molecular role and mechanisms involved are still unknown. In this review, we discuss the findings linking eIF5A and mitochondria, speculate about its role in regulating mitochondrial homeostasis, and highlight its potential as a target in diseases related to energy metabolism.

## 1. The Molecular Function of eIF5A

Although discovered almost 50 years ago, eukaryotic translation initiation factor 5A (eIF5A) is still enigmatic in many aspects. eIF5A is a small, ubiquitous, and essential protein highly conserved across eukaryotes and archaea [1]. It is also very abundant—it is among the 100 most abundant proteins in *Saccharomyces cerevisiae*, with approximately 273,000 copies per cell, which is almost twice the number of ribosomes [2]. Originally classified as a translation initiation factor [3,4], it was subsequently reported that the main roles of eIF5A are to promote the translation elongation of mRNAs at sequences encoding for specific peptide motifs and to assist in termination by stimulating the hydrolysis of peptidyl-tRNA [5,6,7,8,9,10,11].

In most eukaryotes, eIF5A features two isoforms, *TIF5A* and *TIF51B* in yeast, and *EIF5A1* and *EIF5A2* in humans, which share an amino acid sequence identity of more than 90% in each organism and are expressed under different conditions. Here, we refer to the most commonly expressed isoforms, Tif51A in yeast and EIF5A1 in humans, as eIF5A. It is the only known cellular protein containing the unusual and essential amino acid hypusine (Nɛ-(4-amino-2-hydroxybutyl)lysine). Hypusination is critical for eIF5A function and results from a two-step post-translational reaction that requires two enzymes, deoxyhypusine synthase (DHPS) and deoxyhypusine hydroxylase (DOHH) (Figure 1). First, DHPS transfers the aminobutyl moiety from the polyamine spermidine to the ɛ-amino group of a specific lysine residue (Lys51 in yeast and Lys50 in human) to generate an intermediate. Second, DOHH immediately and irreversibly catalyzes the hydroxylation of the deoxyhypusine residue to hypusine, yielding the active hypusinated and mature form of eIF5A [12]. Consequently, intracellular hypusinated eIF5A (hyp-eIF5A) correlates with cellular eIF5A activity. Eukaryotic translation initiation factor 5A can undergo other post-translational modifications, such as acetylation (in Lys47 and Lys68 residues), which is assumed to exclude hypusination [13,14], or phosphorylation (in Ser2) [15], the role of which is not completely understood.

Regarding the structure of eIF5A, different studies, whose subjects range from archaea to humans, have been published in the last decades [16,17,18] showing how hyp-eIF5A folds into a two-domain structure of predominantly β-sheet character, in which the N-terminal portion harbors the unique feature of eIF5A, the hypusine residue [19]. This residue is located at the tip of an extended, unstructured, and exposed loop (hypusine loop) resembling a tRNA. After binding to the already-formed 80S ribosomal complex, hyp-eIF5A is predicted to lie adjacent to the P-site tRNA overlapping the E-site [20,21,22]. In this way, hyp-eIF5A prevents ribosomes from stalling at specific sequences by projecting the hypusine-containing domain toward the P-site to sterically restrict the position of the residue placed on it. Specifically, hyp-eIF5A stimulates the synthesis of proteins by favoring peptide bond formation between critical amino acid residues known to be poor substrates for the reaction, such as stretches of three or more consecutive proline residues (PPP) or polyproline motifs, but also combinations of proline, glycine, and charged amino acids [9,10,11]. Thus, hyp-eIF5A assists in the translation of only a part of the overall mRNA population, which is its distinctive feature. In connection with its main role in translation, hyp-eIF5A can also be localized to the endoplasmic reticulum (ER), where it is associated with ribosomes bound to the ER membrane, and it seems to facilitate the co-translational translocation of some proteins into the ER, such as collagen [23,24,25,26]. Thus, blocking eIF5A hypusination upregulates the stress-induced chaperones in yeast [25] and leads to ER stress in mammalian cells [26,27].

Several studies have indicated that eIF5A is involved in processes that are not directly related to protein synthesis. The structural features of eIF5A suggest it offers the potential to interact with nucleic acids. The C-terminal domain resembles the cold-shock domain (CSD), common in DNA- and RNA-binding proteins, while the N-terminal carries the hypusine residue, which contains two positive charges and resembles spermidine, a molecule known to interact specifically with DNA and RNA. Indeed, hyp-eIF5A has been reported to bind to some RNA molecules in a sequence-specific manner [28,29], and to assist with the transport of newly generated mRNAs from the nucleus to the cytoplasm [28,30]. Moreover, eIF5A mutants exert a considerable impact on the balance between mRNA recruitment to ribosomes for translation and its degradation [24,31,32], suggesting that eIF5A performs a function in the steps of mRNA decay downstream of decapping [24,31]. Archaeal IF5A also plays a role in RNA metabolism as a moonlighting protein that associates with the ribosomes but also exerts RNAse activity [33].

As previously stated, hyp-eIF5A assists in the translation of specific proteins that contain critical motifs in their amino acid sequences, although it is likely that we currently know only a small portion of its direct targets. Thus, the key role that eIF5A plays in different cellular processes is mostly due to the broad spectrum of cellular functions that its direct targets present. One of the major roles of hyp-eIF5A resides in cell proliferation and animal development. Eukaryotic translation initiation factor 5A and its hypusination are essential for cell proliferation in eukaryotes, and the disruption of eIF5A or DHPS genes, as well as the inhibitors of DHPS, cause growth arrest and strong anti-proliferative effects, including apoptosis [13,34,35,36,37,38,39,40,41]. Hyp-eIF5A also mediates efficient autophagy through the translation of the autophagy master transcription factor TFEB and the ATG3 protein, the latter involved in the lipidation of LC3B and formation of the autophagosome [42,43]. Additionally, eIF5A plays an important role in proper cytoskeleton organization and cell shape [44,45,46] through the translation of formins in eukaryotes. In yeast, eIF5A is needed for the translation of the polyproline-containing formin Bni1, which is involved in polarized growth during mating [47]. Accordingly, a mechanistic connection has been demonstrated between eIF5A and diaphanous, the formin involved in actin-rich cable assembly during the embryonic dorsal closure of *Drosophila* and the migration of neural stem cells [48]. Hyp-eIF5A has also been described to promote cell migration, invasion, and metastasis by controlling the expression of a set of key signalling molecules including RhoA and Rho-associated kinase, two cytoskeleton-regulatory proteins involved in promoting cell migration [49], and by directly regulating MYC biosynthesis at specific pausing motifs [50]. Specifically, EIF5A2 isoform has been shown to promote the epithelial-mesenchymal transition in several types of cancer cells [51]. Eukaryotic translation initiation factor 5A has also been implicated in the regulation of apoptosis but the mechanism involved seems tangled given that this function appears to be opposite to the promotion of proliferation [52,53,54]. It was recently found that, in response to stress, hyp-eIF5A promotes the translation of the tumor suppressor and pro-apoptotic factor p53, which contains polyproline motifs sensitive to the action of eIF5A [55] and works as a transcription factor in charge of triggering a variety of antiproliferative programs.

The essential role eIF5A plays in the stated cellular processes implicates this protein in the pathogenesis of a wide variety of human diseases. Increasing evidence suggests that hyp-eIF5A plays an important role in modulating virus propagation. It has been defined as an essential cofactor of the human immunodeficiency virus type 1 (HIV-1) Rev transport factor. Through specific Rev binding, it participates in the translocation of unspliced viral mRNAs across the nuclear envelope [56] and can behave as a nucleocytoplasmic shuttling protein [57]. Although HIV was the first virus suggested to require eIF5A, this factor also participates in the replication of other viruses, such as the Marburg virus (MARV) and Ebola virus [58]. A second human pathogenesis with a well-defined link to eIF5A is diabetes. In mouse models of diabetes, hyp-eIF5A in pancreatic islet β-cells is responsible for the translation of cytokine-induced transcripts, as well as for the activation and proliferation of T helper cells [41,59,60].

The two paralogous genes encoding eIF5A, *EIF5A1* and *EIF5A2*, are expressed under different conditions. *EIF5A1* is ubiquitously expressed in all mammalian tissues and cell types, whereas *EIF5A2* shows restricted expression in healthy tissue (being almost undetectable) but is overexpressed in certain tissues or cancer cells. The overexpression of both eIF5A isoforms has been observed in several tumors and triggers cell migration, invasion, and cancer metastasis (see review [51] for details), but *EIF5A2* is considered a potential oncogene and diagnostic or prognostic marker [61,62] because it is associated with poor survival, advanced disease stage, poor response to chemotherapeutic drugs, and metastasis. Genetic variants of eIF5A genes have been identified as the basis of certain rare neurodevelopmental disorders in humans [63].

The inhibition of the eIF5A function has emerged as a potential target for treating the aforementioned diseases. The inhibition of eIF5A hypusination can be achieved by means of DHPS inhibitors, such as GC7 (N1-guanyl-1,7-diamonheptane), deoxyspergualin, or semapimod; DOHH inhibitors, such as ciclopirox, deferiprone, or mimosine; or inhibitors of ornithine decarboxylase (ODC), such as DFMO (difluoromethylornithine) (Figure 1). DFMO is an irreversible inhibitor of ODC, which is the rate-limiting enzyme of polyamine biosynthesis. Therefore, DFMO acts to reduce polyamine levels and does not specifically inhibit eIF5A hypusination [64]. DFMO has been used to decrease the replication of several RNA viruses, including Ebola, dengue, Zika, polio, and Coxsackievirus [58,65] and in cancer prevention/therapy [66]. Deferiprone and its structural analog, ciclopirox, are used in the treatment of iron overload and fungal infections, respectively. However, all three DOHH inhibitors affect the activity of other enzymes, such as proline hydroxylase enzyme [37]. Among the known DHPS inhibitors, GC7, a diaminoheptane derivative, is the most efficient inhibitor (K_i_ value for GC7, 0.01 µM, compared to K_m_ for spermidine, 4.5 µM) [67] and widely used today to inhibit eIF5A hypusination in mammalian cells [68,69]. There are currently no inhibitors that act directly on eIF5A or, more selectively, on eIF5A2: this is a possible avenue for future research and development.

Lastly, the role of eIF5A in aging has been extensively studied in the last decade. eIF5A is implicated in long-term memory, adaptive immune response, cardiovascular function, and mitochondrial function; the failures of these processes are hallmarks of aging [70].

In this review, we focus on the relationship between eIF5A and mitochondrial metabolism as well as mitochondria-related diseases, with the intention of providing a summary of recent data linking eIF5A to mitochondria in different organisms.

## 2. Mitochondrial Metabolism in Health and Disease

Mitochondria are the main producers of energy in the form of ATP, which is required for key cellular processes. As such, they are essential for eukaryotic life. Mitochondria are derived from the endosymbiosis of α-proteobacteria and host several metabolic pathways, such as the tricarboxylic acid (TCA) cycle, β-oxidation, and lipid synthesis. The TCA cycle and the electron transport chain (ETC) generate ATP from the redox gradient [71]. These bilayer subcellular organelles contain within their own genome (mtDNA) 8 or 13 protein-coding genes (in yeast and human, respectively) that encode for critical proteins mainly implicated in oxidative phosphorylation (OXPHOS) [72]. This genome is replicated and transcribed independently of the nuclear genome, but both genomes must work together to ensure correct cell function. Around 1500 nuclear-encoded proteins are targeted to the mitochondria, which requires a complex import, processing, and assembly system [73]. By processing oxygen to provide energy for cell function, mitochondria have become central players in aerobic life and are critical in many aspects of health, disease, and aging [74,75,76]. When electrons escape as a by-product of oxidative respiration and partially reduce oxygen, mitochondria generate reactive oxygen species (ROS). This happens even in normal conditions of efficient oxygen reduction [77,78]. Under disease conditions, mitochondria become dysfunctional and generally exhibit three main impairments: excess ROS emission, uncoupled OXPHOS, and abnormal Ca^2+^ uptake [79,80]. These defects trigger damage to macromolecules and alterations in the energy supply, the redox environment, mitochondrial signaling, and cell viability. To attenuate these negative effects, mitochondria have developed different quality control pathways to maintain their critical functions and reduce mitochondrial stress. A key quality control pathway is mitophagy, the specific autophagic removal of mitochondria [81]. Moreover, mitochondria show a very dynamic nature through fusion and fission processes, which allows them to adapt to different stresses by remodeling mitochondrial networks [82,83]. Another essential quality control pathway is the response to stress caused by import defects and alteration of lipid metabolism, which consists of the induction of components of the heat shock response and translation attenuation [84]. In addition, mitochondria can sense matrix protein misfolding and induce an adaptive transcriptional program to ensure maintenance of mitochondrial proteostasis [85]. When the cellular damage is too great, mitochondria play an important role in signaling for apoptotic cell death [86].

Mitochondrial function declines during brain aging [87,88,89], but also in aged muscle, heart, liver, and adipose tissues [90]. Thus, in aged cells, there is a reduction in the number and density of mitochondria, as well as in mitochondrial biogenesis [91], ATP production, and respiratory chain capacity/activity [92,93]. Aged cells also show altered mitochondrial dynamics, a decline in mitophagy and mitochondrial quality control systems, and increased mtDNA damage [83,94]. Given its essential role in cells, mitochondrial dysfunction can ultimately affect several biological processes and has emerged as a prominent signature of metabolic, cardiovascular, inflammatory, and neurodegenerative diseases; cancer; and many age-related diseases [95,96,97,98,99]. Because of this, it is important to understand the mechanisms of mitochondrial biology to allow the development of effective treatments.

## 3. The Expression of eIF5A Isoforms Responds Differentially to the Cellular Metabolic State

Most eukaryotes contain two paralogous genes encoding two highly homologous isoforms of eIF5A. These two genes show a clearly differential expression pattern in mammals and yeasts, suggesting a different functional specialization, which, however, has not yet been clearly documented in molecular terms. Most current information about the differential regulation of eIF5A isoforms has been obtained from studies in yeast and indicates the influence of the cellular metabolic and respiratory state.

The adaptation of cellular metabolism to external circumstances is important for most organisms, and especially for yeast, which deals with a continuously changing environment. The expression of eIF5A isoforms shows a pattern of opposite regulation under fermentative and respiratory conditions. Yeast cells tend toward fermentative rather than respiratory metabolism. Although energetically less efficient than respiration, in terms of ATP production, fermentation allows cell activities to proceed at higher rates and enables more competitive growth and survival. This preferred fermentative metabolism is also found in mammalian cancer cells, in which the increase in biomass is prioritized [100].

Glycolysis and fermentation yeast genes are induced in the presence of oxygen and glucose, whereas the genes involved in the use of alternative carbon sources, including respiratory enzymes from the TCA cycle, ETC, and OXPHOS, are subject to glucose repression [101,102,103]. High glucose levels maintain the activity of protein kinase A and target of rapamycin complex 1 (TORC1) signaling pathways, promoting proliferation while inhibiting mitochondrial respiration. Under these conditions, *TIF51A* is constitutively expressed while *TIF51B* is poorly expressed, being almost undetectable. Like other proteins involved in translation [104,105,106], Tif51A is highly active and positively regulated by TORC1 to couple biosynthetic activity to the abundant nutrient availability [107]. After glucose becomes limiting and with sufficient oxygen, yeast cells switch their metabolism to aerobic respiration. During this transition, the expression of *TIF51A* is increased two to fourfold [107], as is that of genes involved in the TCA cycle, ETC, and OXPHOS [101,102,103,108,109], while *TIF51B* expression is continuously decreased. As the glucose concentration drops, TORC1 is inactivated, leading to a slow reduction in the translation and synthesis of ribosomal components [104,105,106]. This means that cell growth is slow and less cytoplasmic translation is needed, but, surprisingly, more eIF5A protein is demanded [107]. Accordingly, upon exponential growth under non-fermentative conditions, such as glycerol or ethanol, *TIF51A* mRNA levels are also significantly increased compared to the levels during exponential growth in glucose, whereas *TIF51B* levels are downregulated [107].

The main factors involved in yeast metabolic reprogramming between the two alternative physiological states, fermentation and respiration, are protein kinase A, Snf1, and the heme/oxygen responsive transcription factors Hap1 and the Hap2/3/4/5 complex. In the transition, Hap1 and Hap4 are induced and upregulate genes involved in respiratory processes, such as the TCA cycle, ETC, and OXPHOS [101,110,111,112,113]. Hap1 is also the transcription factor involved in the upregulation of *TIF51A* expression after the metabolic shift to respiratory growth; this regulation is lost in a *hap1* mutant [107]. Hap1 responds to the increase in heme cellular levels caused by the augmented metabolic flux into the TCA cycle produced under respiratory conditions [108,109,111,114]. Remarkably, the genetic regulation of eIF5A is clearly different from that of other translation factors. The expression of most translation factors decreases after this metabolic shift, but Tif51A shows a unique and dual regulation with an initial reduction caused by TORC1 inactivation and a subsequent progressive increase through the action of Hap1 [107]. This clearly highlights the essential role of eIF5A in the respiratory process.

By contrast, high oxygen/heme levels lead to *TIF51B* repression through the synergistic action of the two DNA-binding repressor proteins Rox1 and Mot3 [115,116,117,118], with Rox1 activated by heme-bound Hap1 [110]. However, under hypoxic conditions and reduced heme and iron levels, Tif51A protein expression drops. The mechanism of this negative regulation is suggested to be the combination of a decrease in `0the activity of DOHH, which uses oxygen as a substrate in eIF5A hydroxylation [119] and the action of Hap1, which can also act as a repressor [107]. On the other hand, Hap1 acting as a repressor under hypoxic conditions downregulates *ROX1*, which induces *TIF51B* expression [120].

In yeast, the control of both eIF5A isoforms by Hap1 through the activation/repression of *TIF51A* expression and the Rox1-mediated repression/activation of *TIF51B* allows opposite regulation of two genes by only one transcription factor. Thus, this differential expression affects different metabolic outcomes, with Tif51A promoting respiration and Tif51B promoting anaerobic glycolysis. It should be noted that yeast Hap1 protein features no homologs in mammalian cells, but the existence of another non-homologous transcription factor mediating a similar regulation of the *EIF5A1* and *EIF5A2* human isoforms cannot be ruled out. Indeed, there are examples of different expressions of eIF5A human isoforms connected to different metabolic outputs. In human hepatocellular carcinoma (hHCC) patient samples, which usually show reprogramming of intracellular metabolism, *EIF5A2* was upregulated. Moreover, the ectopic expression of *EIF5A2* in hHCC cells increased the expression of glycolysis enzymes together with lactate dehydrogenase, promoting anaerobic glycolysis [121]. Thereby, glucose uptake and lactate secretion were increased through the upregulation of glycolytic enzymes, which is the most common metabolic reprogramming of most cancer cells [122].

Additionally, eIF5A has been identified as essential for the expression of mammalian hypoxia inducible factor 1α (HIF-1α), the master regulatory transcription factor of the cellular adaptive response to hypoxia [119]. Eukaryotic translation initiation factor 5A in its acetylated form, which is inactive, increases under long hypoxic periods and is responsible for the decrease in HIF-1α activity. Although the mechanism underlying eIF5A and HIF-1α expression is yet to be elucidated, this regulation makes eIF5A an attractive therapeutic target because HIF-1α mediates the adaptive response in the hypoxic environment of tumor spheroids [119,123].

In summary, data from yeast and humans support the differential expression of *EIF5A1* and *EIF5A2* genes linked to the metabolic state of cells, although it is not clear whether this differential expression is the cause or consequence of the metabolic cellular status. We want to stress that many studies investigating the eIF5A function in mammalian models use the hypusine inhibitor GC7, which, to date, is believed to reduce the hypusination of both eIF5A isoforms. If each eIF5A isoform promotes a different type of metabolism, that is, aerobic or anaerobic glycolysis, results inhibiting the hypusination of both isoforms simultaneously are more difficult to interpret.

## 4. Links between eIF5A and Mitochondrial Function

There has long been evidence of connections between eIF5A, its hypusination enzymes and the hypusination substrate spermidine, and mitochondrial function, but the sum of all the data still provides a picture that is not entirely clear and sometimes seems contradictory.

On the one hand, different reports have documented that both a defect in and an excess of eIF5A create deleterious effects on mitochondrial function. For example, in rat cardiac muscle cells, the overexpression of eIF5A induced by the antitumor agent doxorubicin resulted in a gradual increase in ROS and an increase in Ca^2+^ influx in mitochondria [54]. These changes correlated with a loss of mitochondrial transmembrane potential and the induction of apoptosis, whereas the disruption of *EIF5A1* expression reduced apoptosis. These results were in good agreement with results reported by Sun et al. (2010) [124], which were obtained in human cells in which *EIF5A1* overexpression was stimulated by viral infection, but also the overexpression of a *EIF5A1* mutant incapable of being hypusinated induced apoptotic cell death through the intrinsic mitochondrial pathway. The increase in *EIF5A1* levels caused a loss of mitochondrial membrane potential and resulted in the translocation of the apoptotic marker B-cell lymphoma 2-associated X (Bax) protein to the mitochondria, the release of cytochrome c, and caspase activation. A further interesting aspect of this study was that a proteomic analysis of HeLa cells with eIF5A overexpression showed the upregulation of mitochondrial proteins [124]. Therefore, in these studies, the dysregulation of hyp-eIF5A and non-hypusinated eIF5A protein levels provoked mitochondrial dysfunction and apoptosis.

In the case of yeast, as referred to above, it was identified early that the eIF5A isoform encoded by the *S. cerevisiae TIF51A* gene is required for growth in the presence of oxygen, while the isoform encoded by the *TIF51B* gene is induced in hypoxic conditions [1]. Indeed, a reduction in Tif51A protein causes the mitochondrial respiration rate to drop [107]. In the fission yeast *Schizosaccharomyces pombe*, a point mutation in the *DOHH* gene (encoded by the *MMD1* gene) causes temperature-sensitive growth and defects in mitochondrial morphology and distribution. At a non-permissive temperature, microtubules, which mediate mitochondrial positioning, display aberrant organization and mitochondria aggregated at the two cell ends [125].

Recent studies in different disease contexts, such as kidney transplantation, stroke, and malaria infection, have highlighted the positive connection between the activities of eIF5A and mitochondria, pointing to the inhibition of hyp-eIF5A as a way to protect cells under transitory situations of low oxygen availability that otherwise contribute to mitochondrial damage and cell death. Melis et al. (2017) [126] investigated the possible link between eIF5A hypusination and cellular resistance to hypoxia/anoxia. They showed that treatment with GC7- or RNA interference-mediated inhibition of DHPS or DOHH prevented anoxia-induced cell death in mouse renal cells. Importantly, GC7 treatment induced a reversible metabolic shift toward glycolysis that was accompanied by mitochondrial remodeling and the downregulation of the expression and activity of the ETC respiratory chain complexes together with decreased mitochondrial oxygen consumption rate and attenuated anoxia-induced generation of ROS. Together, these data show that reduced hyp-eIF5A activity leads to mitochondrial silencing. This was confirmed in rat ischemia-induced renal injury and pig kidney transplantation models, which demonstrated the beneficial effects of mitochondrial silencing through hyp-eIF5A inhibition to prevent anoxia-induced cell death [126]. More recently, the same group used a pig transplantation model to show that eIF5A inhibition by GC7 treatment preconditioned kidneys for transplantation from donors with brain death, which was accompanied by reduced oxygen intake [127]. Specifically, GC7 treatment appears to preserve antioxidant defenses by increasing the expression of mitochondria-protecting proteins (superoxide dismutase, heme oxygenase, and others) and mitochondrial integrity/homeostasis by decreasing dynamin-related protein 1 expression and increasing mitofusin-2 expression. These protective effects resulted in a better transplantation outcome [127]. The preservation of mitochondrial function by reducing its activity during hypoxic conditions via hyp-eIF5A inhibition was demonstrated again by the same group working with a mouse cerebral ischemia model. The loss of mitochondrial membrane potential is a hallmark of neuronal cell death linked to mitochondrial dysfunction that occurs with excessive ROS production, Ca^2+^ release from the mitochondria, and a decrease in internal ATP. GC7 treatment reduced these three effects in neurons treated with depolarization agents, preserving mitochondrial membrane potential. In vivo, GC7 reduced infarct volume and post-stroke cognitive deficits in mice [128]. In a different context, studies on infants with malaria infection have shown the parallel occurrence of cellular hypoxia that results in apoptosis of cardiac ventricular myocytes. An in vitro model of malaria infection in human cardiomyocytes determined that reduction of hyp-eIF5A levels by GC7 treatment leads to a decrease in the release of cytochrome c and lactate from damaged mitochondria and reduces proinflammatory and pro-apoptotic myocardial caspase-1 activity. These results demonstrate that the administration of GC7 in a malaria-simulating in vitro model prevents cardiac damage driven by hypoxia [129].

All the previous studies on different models and organisms clearly demonstrate the positive role of hypusinated eIF5A in promoting mitochondrial respiration and function, but also indicate that an excess of eIF5A, hypusinated or not, deregulates mitochondrial functioning. The mechanisms underlying this eIF5A–mitochondria relationship are only beginning to be elucidated and are discussed in the following sections.

## 5. Polyamines Regulate Mitochondrial Function: eIF5A-Dependent and Independent Effects

Mammalian cellular polyamine levels are tightly controlled and it has long been known that polyamines (spermidine, spermine, and their precursor putrescine) perform regulatory functions in the mitochondria [130,131,132]. The depletion of polyamines provokes oxidative stress and induces the mitochondrial permeability transition that ultimately conducts cells to apoptosis or necrosis [133]. By contrast, the addition of spermine to depolarized mitochondria restores transmembrane potential [134]. Polyamine effects on mitochondria may be executed through eIF5A, given that one of the main roles of spermidine is as a substrate for eIF5A hypusination [12,130]. However, it seems that other eIF5A-independent polyamine roles may target mitochondrial function directly. For example, it has been found that spermidine promotes ribosomal translation initiation of yeast *COX4*, one of the subunits of cytochrome c oxidase (complex IV) in which a defect compromises mitochondrial respiration [135]. The initiation of *COX4* mRNA translation is upregulated by spermidine through ribosome shunting, an unconventional translation initiation mode also used by viruses and a few eukaryotic mRNAs mediated by an extended hairpin structure in the 5′ untranslated region [136]. Although eIF5A has not been investigated in relation to *COX4* translation, it has been documented to promote other unconventional modes of translation initiation during virus infection [137] and for specific cytoplasmic mRNAs [138,139]. A profound comprehension of the mechanism through which polyamines affect mitochondrial function will make possible to discern between eIF5A-mediated and independent effects.

## 6. Eukaryotic Translation Initiation Factor 5A’s Subcellular Localization and Its Association with Mitochondria

Eukaryotic translation initiation factor 5A is a very abundant protein that is mostly localized in the cytoplasm. However, other non-cytoplasmic localizations have been described, including the mitochondria, although the quantitative and functional relevance of these alternative subcellular localizations are still unclear.

As discussed previously, one of the subcellular localizations of eIF5A is the ER membrane, where it seems to mediate co-translational translocation of proteins into the ER [23,25,26]. Eukaryotic translation initiation factor 5A has also been detected in the nucleus, which it can enter through the nuclear pore complexes owing to its small size [140]. Nuclear exportins for eIF5A have been found in mammals (Xpo4) and in yeast (Pdr6) [30,140]. The nuclear localization of eIF5A seems to be regulated by its reversible acetylation, a modification that seems to exclude hypusination (reviewed in [141]). Thus, the current consensus is that the unmodified eIF5A is distributed throughout the cell, whereas acetylated eIF5A accumulates in the nucleus and hyp-eIF5A in the cytoplasm. The nucleus–cytoplasm shuttling of eIF5A has been proposed to facilitate the nuclear export of specific mRNAs and proteins, although this function is not clearly understood [141].

It has also been proposed that the nuclear export of eIF5A is mediated by Xpo1/Crm1, although later results argue against Xpo1 as a direct eIF5A exportin [24,57,142]. Interestingly, the inhibition of Xpo1 promotes the accumulation of eIF5A protein in the mitochondria in a human ovarian cancer cell line, where it induces apoptosis [143]. It was discovered that eIF5A interacts with the insulin-like growth factor 2 mRNA-binding protein (IGF2BP1) in the cytoplasm, preventing eIF5A accumulation in the mitochondria. Xpo1 is the nuclear exportin of IGF2BP1; thus, the inhibition of the Xpo1-mediated nuclear export of IGFBP1 results in localization of eIF5A in the mitochondria. IGF2BP1 therefore acts as a regulator of eIF5A’s localization and pro-apoptotic function in the mitochondria [143]. Other reports have also proposed that eIF5A is associated with the mitochondria. In a proteomic study to determine differentially expressed mitochondrial proteins in a metastatic compared to a non-metastatic nasopharyngeal carcinoma cell line, eIF5A was one of the most induced proteins from purified mitochondria, together with proteins involved in mitochondrial redox metabolism, respiratory electron transport, and mitochondrial membrane potential [144].

During the translation of some mRNA transcript variants of the *EIF5A1* human gene, the use of an alternative start codon gives rise to an eIF5A isoform with an extended 30 amino acids at the N-terminal peptide sequence [145]. This longer eIF5A isoform is much less efficiently translated than the canonical eIF5A, but it is also susceptible to modification by hypusination. The extended N-terminal sequence contains a putative mitochondrial localization signal and, indeed, when the longer eIF5A isoform was overexpressed in human HeLa cells, it was co-purified with the mitochondria [145]. More recently, the same group investigated the role of the longer eIF5A isoform on mitochondrial function [146]. They used a specific siRNA to deplete only the longer eIF5A isoform without affecting the canonical one and observed the downregulation of the mRNA levels of several genes involved in mitochondria biogenesis, as well as a reduction in the levels of several OXPHOS proteins in HeLa cells. Contrary to the results of depleting canonical eIF5A, the depletion of the N-terminal extended eIF5A isoform led to an increase in oxygen consumption; however, it also produced more ROS and mitochondrial fragmentation, and increased the expression of the BAK pro-apoptotic protein, suggesting that the longer eIF5A isoform is necessary for mitochondrial dynamics and that its depletion results in mitochondrial dysfunction and apoptosis [146]. Except for the increase in oxygen consumption, other effects of depleting the longer eIF5A isoform recapitulate those found when depleting or inhibiting the canonical mammalian eIF5A isoform, raising the possibility that the strategies used, such as hypusination inhibition or RNA interference, against the canonical EIF5A1 sequence could also act on the longer eIF5A isoform. Thus, the results observed may be a consequence of a lack of the less-expressed isoform. This possibility cannot currently be ruled out and further work is needed to clarify this point.

From the works cited above, it may be inferred that the role of eIF5A in mitochondria is directly linked to its association with this organelle, but this is still uncertain.

## 7. Eukaryotic Translation Initiation Factor 5A’s Molecular Roles in Mitochondrial Function

The studies reviewed in the previous sections demonstrate that eIF5A is necessary for correct mitochondrial function; however, the results are still puzzling. Here, we review very recent works that propose different molecular mechanisms through which eIF5A may affect mitochondrial performance.

The results presented by Puleston et al. (2019) [147] are revealing about the role of eIF5A in mitochondria. First, the authors showed that the inhibition of hyp-eIF5A limits mitochondrial OXPHOS in mammalian cells and that hyp-eIF5A is necessary for increasing respiration under conditions of limited glycolysis. Furthermore, the authors demonstrated in an OXPHOS-dependent mouse macrophage activation context that the inhibition of hyp-eIF5A reduced the activity of the TCA cycle and reduced the expression of many mitochondrial proteins at the protein level without affecting mRNA levels. These results are in agreement with previous results reported by Melis et al. (2017) [126]. The eIF5A-sensitive mitochondrial proteins identified by Puleston et al. (2019) [147] included ETC complex components, some TCA enzymes (e.g., succinyl-CoA synthetase and succinate dehydrogenase), and TCA-feeding enzymes (e.g., pyruvate dehydrogenase). However, other TCA proteins were less affected (e.g., citrate synthase and isocitrate dehydrogenase) and glycolysis enzymes were unaffected, indicating the specificity of the effect. The authors investigated the possibility that the translation of specific mitochondrial targeting signals (MTSs) was dependent on hyp-eIF5A. They showed that the MTSs of some hyp-eIF5A-sensitive proteins were sufficient to confer hyp-eIF5A-dependent translation efficiency when fused to reporter proteins. Puleston et al. (2019) [147] concluded that hyp-eIF5A regulates mitochondrial respiration, at least in part, by promoting the translation of the MTSs of some mitochondrial proteins, an essential step when the demands of OXPHOs increase, such as during macrophage activation. Although MTS sequences contain stretches of repetitive amino acids and are rich in charged amino acids that may slow translation, it is not clear, as the authors state, how hyp-eIF5A impacts the translation efficiency of some of these MTSs, but not that of others with similar global characteristics.

New investigations into the protective role of eIF5A inhibition during kidney ischemia have detailed the metabolic transition from aerobic oxidative phosphorylation toward anaerobic glycolysis upon GC7 treatment and have shown that this treatment regulates the expression of glucose transporters [148]. In proximal renal cells under eIF5A hypusination inhibition, oxygen consumption decreased but glucose consumption and lactate efflux increased, as well as the dependence on glucose import and glycolysis, demonstrating a metabolic shift to exclusive anaerobic glycolysis. During in vitro and in vivo GC7 inhibition, the glucose efflux from renal proximal cells was impaired through the repression of the facilitated glucose transporter GLUT1, thereby increasing the availability of glucose in the proximal cells. These results explain the survival of kidney cells under hypoxia, with energetic demands being met by a shift to anaerobic glycolysis. However, the molecular reasons for this metabolic change are unclear, given that GLUT1 does not contain putative eIF5A-dependent peptide motifs. It is also unknown whether the increase in intracellular glucose levels resulting from the inhibition of GLUT1 is a cause or consequence of this GC7-mediated metabolic change [148].

Another recent study has established a molecular connection between eIF5A and the mitochondrial fusion process [149]. Its authors investigated the role of the cardiovascular remodeling transcription factor Krüppel-like factor 5 (Klf5) in vascular senescence and found that Klf5 directly binds to the *EIF5A* promoter and activates its transcription to preserve mitochondria integrity. The modulation of eIF5A by Klf5 concomitantly modulated ATP content, ROS production, and mitochondrial dynamics. Importantly, eIF5A physically interacted with mitofusin 1 (Mnf1), a transmembrane protein of the outer mitochondrial membrane that is a key regulator of mitochondrial fusion and integrity. Eukaryotic translation initiation factor 5A and Mnf1 co-localized in the mitochondria and facilitated the formation of networks of fused mitochondria. By contrast, eIF5A downregulation by Klf5 deficiency or during vascular senescence resulted in mitochondrial fission and led to vascular disease [149]. Although the precise mechanisms through which mitochondrial integrity is maintained through the interaction of eIF5A with Mnf1 are still to be elucidated, it is known that balanced mitochondrial fusion and fission coordinates not only mitochondrial shape, size, and number, but also energy metabolism, the cell cycle, mitophagy, and apoptosis [150].

As referred to in previous sections, it is well known that mitochondrial function declines with age [83]. In the last few years, different in vitro and in vivo models have been used to show a strong connection between spermidine supplementation and life extension in model organisms, and retarded aging in model organisms and humans. This connection has recently been investigated and spermidine effects in aging are mostly mediated by spermidine-induced eIF5A hypusination, which preserves mitochondrial function.

A decrease in polyamines has long been documented as occurring in mammalian cell cultures and human organs during aging [151,152]. In 2009, work by Eisenberg et al. (2009) [153] showed that the exogenous addition of spermidine extended lifespan in yeast, fly, worm, and human cells. Spermidine also reduced age-related oxidative stress in mice. By contrast, polyamine depletion decreased yeast lifespan and increased ROS production and necrosis. The authors observed a correlation between spermidine-induced longevity and the hypoacetylation of histone H3, which led to the upregulation of the autophagy genes *ATG7*, *ATG11*, and *ATG15* and promoted autophagy in all the model organisms tested, which was shown to be crucial for polyamine-enhanced longevity [153].

The beneficial effects of spermidine supplementation on aging were also reported in different organisms, in relation to different age-related aspects connected to the promotion of autophagy [154,155], but also independently [156]. Spermidine was proven to ameliorate cardiovascular aging, improve memory, and reduce cancer mortality, among other beneficial effects [157]. The mediation of spermidine effects through induced hypusination of eIF5A has recently been shown in studies of B-cell immunity in older adults. Spermidine-promoted eIF5A hypusination restored B-cell immunity in old mice through the promotion of autophagy [43]. Hyp-eIF5A maintained autophagy through the translation of the autophagy transcription factor TFEB, which contains polyproline motifs in its amino acid sequence. During aging, there is a decline in TFEB levels, together with those of hyp-eIF5A and spermidine, leading to failures in the immune system [43]. Intriguingly, a proteomic analysis conducted in this study with primary B-cells treated with GC7 showed a reduced expression of TFEB, but not of the previously shown mammalian and *Caenorhabditis elegans* autophagy eIF5A target ATG3 [42], suggesting the influence of the cellular context. Both ATG3 and TFEB contain polyproline motifs susceptible to eIF5A dependency and provide a mechanistic connection between hyp-eIF5A and autophagy, although the presence of polyproline motifs alone in the protein sequence does not seem sufficient in all cases to cause reduced protein levels by inhibiting hyp-eIF5A [43,158].

The mechanistic connections between the positive effects of spermidine on reducing age-related symptoms and the role of hyp-eIF5A in maintaining functional mitochondria have been explored in very recent studies. Schroeder et al. (2021) [159] studied aged mice and found that dietary spermidine passes the blood–brain barrier, increasing hippocampal eIF5A hypusination. Aged mice fed with spermidine showed improvements on several cognitive tests. The authors also showed higher mitochondrial respiration in the mouse brain and in flies, although results in mice were sex- and age-dependent. It has previously been proposed that autophagy is crucial for mitochondrial quality control during aging and neurodegeneration [160]. Accordingly, in *Drosophila*, the downregulation of the essential autophagy gene Atg7, the mitophagy-associated PTEN-induced putative kinase (Pink1), and the homolog of human E3 ubiquitin ligase Parkin (Park) eliminated the spermidine-mediated improvement in respiration [159]. These results are in agreement with results from [161] in *C. elegans*, where spermidine inhibition of neurodegeneration and aging was dependent on PINK1 and the worm Parkin ortholog PDR1, which mediate mitophagy.

The beneficial effects of spermidine on mitochondrial and brain cognition health may be mediated by the effector eIF5A [159,162]. Aging *Drosophila* under spermidine supplementation contained higher levels of proteins involved in OXPHOS, but not higher levels of the corresponding mRNA, and this correlated with higher maximal respiration and mitochondrial abundance in brains. In aged *Drosophila* brains, spermidine and hyp-eIF5A levels dropped, and both could be boosted by polyamine supplementation until mid-age, although not in very old *Drosophila* brain. Different genetic approaches for partial reduction of eIF5A hypusination in the fly brain led to reduced mitochondrial respiration, and quantitative proteomic analysis indicated downregulation of mitochondria proteins, specifically OXPHOS proteins. Interestingly, most of the positive effects on mitochondrial function caused by spermidine supplementation were abolished by reducing hyp-eIF5A in fly brains. Liang et al. (2021) [162] also investigated whether eIF5A hypusination was the reason for the previously mentioned extended lifespan obtained in flies, mice, and other organisms with the addition of spermidine [153,163,164]. They observed that life extension was abrogated in hypusination-deficient flies [162]. Finally, spermidine ameliorated the decline in the locomotive and memory functions of aged flies, while the attenuation of eIF5A hypusination enhanced these age-deleterious effects and, importantly, the positive effects of spermidine on locomotion and memory were mostly lost in eIF5A hypusine-attenuated animals. These latter results are in agreement with another recent study in which a reduction of deoxyhypusine synthase activity was associated with a neurodevelopmental disorder in humans [165]. Although the results reported by Liang et al. (2021) [162] reinforced the molecular connection between the beneficial effects of spermidine on aged mitochondria performance and brain functions with eIF5A hypusination, and Schroeder et al. (2021) [159] pointed to a role of hyp-eIF5A in maintaining mitochondrial quality control by autophagy/mitophagy, the mechanistic molecular details are still missing and it is not completely understood how autophagy/mitophagy directly benefits mitochondrial performance.

## 8. Perspectives on the Identification of Mitochondrial Processes and Targets under the Control of eIF5A

The main molecular role attributed to eIF5A is facilitating the translation of specific subsets of proteins containing eIF5A-dependent motifs [10,11]. However, other less-characterized molecular roles have been proposed relating to the capacity of eIF5A to bind RNAs and regulate their metabolism. From the above results, it is clear that proper levels of hyp-eIF5A are necessary to preserve OXPHOS and mitochondrial function. The studies described above have proposed that eIF5A might be involved in the translation of specific mitochondrial proteins containing eIF5A-dependent MTS, in regulating the flux of metabolites to mitochondria to maintain aerobic metabolism through the regulation of glucose transporters, in sustaining mitochondrial dynamics through the interaction with proteins involved in mitochondrial fusion (Mnf1), in promoting autophagy by facilitating the synthesis of ATG3 and TFEB, or specifically in mitophagy through a mechanism that is dependent on PINK1 and Park proteins. To discern and prove these roles it will be necessary to identify the specific molecular process and/or the specific protein targets and conditions under the direct control of eIF5A.

In an attempt to identify mitochondrial protein candidates for direct eIF5A-dependent translation, we searched for peptide motifs causing eIF5A-dependent ribosome stalling [11] in the *S. cerevisiae* mitochondrial proteome. We found that eIF5A motif abundance in the 1117 nuclear-encoded mitochondrial proteins is only slightly higher than that in total yeast proteins, but these motifs are scarcely represented in the 10 mitochondrial DNA-encoded proteins (Figure 2). Proteins involved in the TCA cycle show an average of 3.3 motifs/protein, higher than the average in the total yeast proteome (2.8 motifs/protein) and much higher than that in OXPHOS proteins (1.8 motifs/protein). However, the most represented eIF5A-dependent motifs were the same (GGA, GGG, and KPG) in TCA and OXPHOS proteins, with only two PPP motifs: one in the succinyl-CoA ligase (Lsc1) TCA enzyme, and another in the subunit of succinate dehydrogenase (Sdh4), which participates in the TCA cycle and OXPHOS.

A search for yeast mitochondrial proteins with longer polyproline motifs showed that Yta12, Srv2, and Tim50 contain stretches of up to nine, six, and seven consecutive prolines, respectively (Figure 2). Yta12 (homologous to human AFG3L2) is part of the conserved mitochondrial m-AAA protease, which is composed of the Afg3 and Yta12 proteins, and located at the inner mitochondrial membrane (Figure 3). Yta12/Afg3 regulates mitochondrial proteostasis by mediating protein maturation and degradation and is required for the correct assembly of mitochondrial enzyme complexes [166]. Interestingly, the Yta12/Afg3 complex is also involved in the splicing of the mitochondrial mRNAs containing introns *COX1* and *COB*, which encode subunit 1 of cytochrome c oxidase and cytochrome b, respectively, which are part of the ETC. Thus, deficiency in Yta12/Yta10 causes low respiration due to, among other effects, deficient ETC assembly [167]. Srv2 (homolog to human CAP1 and CAP2) mediates actin assembly at mitochondria, and deletion of Srv2 causes elongated hyperfused mitochondria and reduces respiration. Interestingly, Srv2 interacts with mitochondrial fission GTPase Dnm1/DRP1 [168]. Finally, Tim50 (a homolog of human TIMM50) is an essential subunit of the mitochondrial inner membrane TIM23 complex, which mediates the import of the majority of mitochondrial proteins through recognition of their MTSs (Figure 3) [169]. Although these three proteins are tempting candidates for mediating the effects of eIF5A on mitochondrial function, the presence of polyproline stretches is not a sufficient requirement to create dependence on eIF5A [43,158]. Future studies will determine the precise mechanistic connections between eIF5A and the already described hyp-eIF5A-sensitive mitochondrial proteins and will identify new mitochondrial target proteins and processes under the control of eIF5A (Figure 3).

## Figures and Tables

**Figure 1 ijms-23-01284-f001:**
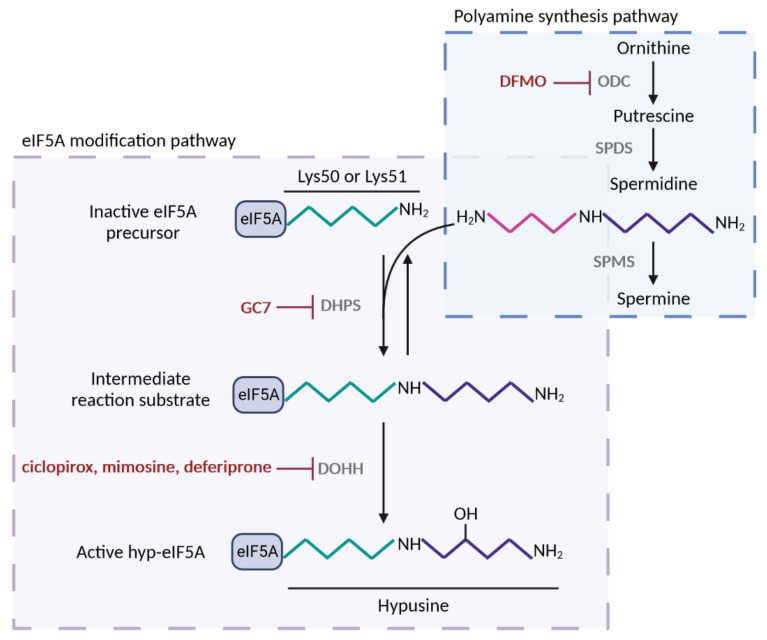
Polyamine-hypusine pathway and its pharmacological inhibitors. Spermidine substrate for eIF5A hypusination is obtained by the conversion of the polyamine ornithine in putrescine by the enzyme ornithine decarboxylase (ODC); next, spermidine is synthesized from putrescine by spermidine synthase (SPDS). Alternatively, spermidine is converted in spermine by spermine synthase (SPMS). Hypusine modification of lysine-50 (human) or lysine-51 (yeast) residue of eIF5A occurs by the addition of spermidine via two consecutive enzymatic reactions. First, deoxyhypusine synthase (DHPS) transfers the aminobutyl group of spermidine to the amino group of lysine generating an intermediate substrate, which does not accumulate. Second, deoxyhypusine hydroxylase (DOHH) adds a hydroxyl group and forms the hypusine residue of eIF5A, which confers the activity to the protein. eIF5A post-translational modification can be suppressed by inhibitors of DHPS and DOHH, but also by inhibition of ODC, the rate limiting enzyme for spermidine biosynthesis. Figure processing was carried out using BioRender software.

**Figure 2 ijms-23-01284-f002:**
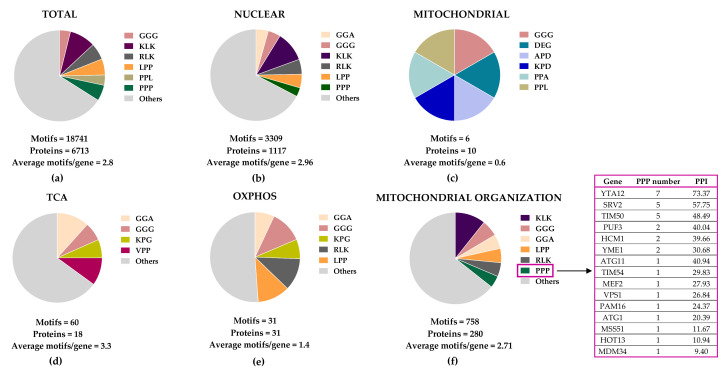
Distribution of eIF5A-dependent motifs in *Saccharomyces cerevisiae* mitochondrial proteins. Distribution of the 43 highest-scoring eIF5A-dependent ribosome-pausing tri-peptide motifs [11] in the proteins of the whole yeast genome (**a**), nuclear-encoded mitochondrial proteins (**b**), mitochondrial-encoded proteins (**c**), the tricarboxylic acid (TCA) cycle (**d**), oxidative phosphorylation (OXPHOS) (**e**), and in the mitochondrial organization Gene Ontology functional category (**f**). The table shows the proteins involved in mitochondrial organization with at least one PPP motif. The protein pause index (PPI) is calculated as the sum of the quantitative value of the ribosome pause provoked by depletion of eIF5A in each of the 43 highest eIF5A-dependent tri-peptide motifs [11] found in the amino acid sequence of each protein, and is higher in proteins that are putatively more dependent on eIF5A for their translation.

**Figure 3 ijms-23-01284-f003:**
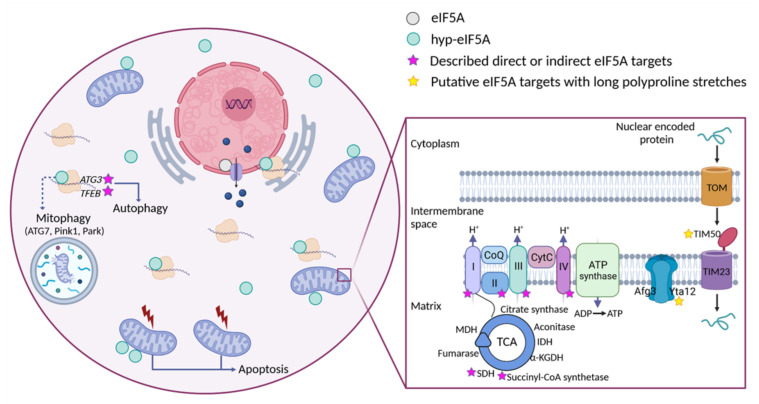
Cellular funtions of eIF5A and model for its role in maintaining mitochondrial activity. eIF5A is known to be implicated in different cellular processes, although the most relevant and mitochondrial-related of these are represented in the Figure. Bound to ribosomes, hyp-eIF5A facilitates translation elongation at specific motifs [10,11], as well as ER-coupled translation [23,25,26]. In the nucleus, eIF5A helps to export certain mRNAs and proteins [141]. eIF5A plays a controversial role in apoptosis, as it has been defined to be necessary to induce the mitochondrial mediated apoptosis [124,146], but also to lead to cell death when inhibited [133]. Hyp-eIF5A promotes autophagy through the translation of the autophagy factors ATG3 (autophagy-related 3) and TFEB (transcription factor EB) [42,43]. Increasing evidence shows a direct link between hyp-eIF5A and mitochondrial function. In addition to its association with mitochondria [143,144,145,146], some proteins of both the TCA and ETC have been described to be directly or indirectly affected under hyp-eIF5A inhibition [126,147]. It has also been proposed that hyp-eIF5A could mediate mitophagy through ATG7 (Autophagy Related 7), Pink1 (the mitophagy-associated PTEN-induced putative kinase), and Park (the E3 ubiquitin ligase Parkin) proteins [170]. Other proteins involved in the mitochondrial transport of nuclear-encoded proteins and mitochondrial organization are considered as putative eIF5A targets (Figure 2). Among these, the mitochondrial inner-membrane integral proteins Yta12 (protease of the Yta12/Afg3 complex and yeast homolog of human AFG3L2), and Tim50 (essential subunit of the TIM23 complex and yeast homolog of human TIMM50) contain long polyproline stretches in their amino acids sequences, suggesting a possible dependence on eIF5A for their translation and, thus, a possible link between hyp-eIF5A and mitochondrial function. Figure processing was carried out using BioRender software.

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
