# Peer review of "Role of eIF5A in Mitochondrial Function"

_ijms, 2022, doi:10.3390/ijms23031284_

Round 1

Reviewer 1 Report

Altogether the review MS is well written and bring a complete overview concerning the relation between eif5A and mitochondria function. MS is perfectly written, and all paragraphs are following a logic sequence. However, some points must be clarify/modify (or corrected) to increase the quality of the paper.

Main concerns

The authors have to take care about some the terms used that bring some confusion: “disruption of eIF5A” “Pharmacologic inhibition of eIF5A activity”,  “inhibition of eIF5A” most of this term refers to an inhibition of DHPS that reduce the quantity of hypusinated form of eif5a. Please don t make a confusion between native eif5a protein and the hypusinated form or eif5a (please use “hyp-eif5a“ when necessary). Please correct accordingly everywhere in the MS.

Eif5A1 versus EIF5A2: throughout the MS these terms are confusing because (as they state when they speak about eif5a is referring to eif5A1) but sometimes they mentioned eif5A1…please harmonized. Similarly most publications concerning eif5A2 were done on cancer cells harbouring a specific metabolic condition (mainly glycolysis). This is mentioned is a paragraph but it is not precise anymore in the rest of the MS.

Page 6: concerning Ref 121,  in this study what is increase is the expression of eif5a not the hypusinated form directly. This has to be mentioned. It might be possible and even suggested that eif5a alone (not hypusinated) has also a role and a function.

Minor points :

Chapter 1: eif5a molecular function and disease

Very interesting chapter but the authors start of this chapter with description of Eif5A pathways in yeast and the consequence in term of modification of cellular functions such as synthesis of PPP peptide bond or the stabilisation of RNA (this chapter has nothing to do with disease neither with mitochondria). Please named this chapter “eif5a molecular function” describing the pure role of the eif5a pathway in eucaryote and in yeast such as an introduction (it would also of interest to give a schema showing the different step leading to hypusinated eif5a (starting from ornithine for example). This should be link to the description of target of the drugs that is not clearly defined (p3 line 134). It would be of interest to give a scheme with the cascade of enzyme leading to hypusinated eif5A and their “specific” inhibitors. DFMO is acting very upstream and is clearly not specific as well as ciclopirox for eif5A and its post translational modification. Similarly ciclopirox is a metal ions chelator acting on a wide variety of enzyme and has a bad specificity on eif5a hypusination inhibition. Only GC7 seems to be quite specific but with a low IC50 (some µM). Please be more precise and give more details about the drugs that are used in the literature to target eif5a hypusination pathway.

Chapter 4

The paragraph concerning polyamines should be separate from this eif5a paragraph. It is 2 different concept and pathways (even if they are link somehow see for example MS from Kazuhiro Nishimura Biochem J, 2005 doi: 10.1042/BJ20041477). It is difficult (for the moment) to discriminate precisely between the action of the modulation of metabolism of the polyamines (putrescine, …) and the direct activity of hypunisated eif5A.

Fig 2 : it is very difficult to interpret this figure. Please give more detail. Where is acting hyp-eif5a ? at the cellular level ? which target did you identify for nuclear effect, peptide prot synthesis,…. ? please mentioned hyp-eif5A instead of eif5A it is confusing. To my knowledge complex III seems to be also a target of hyp-eif5a (melis et al 2017). Please define what is mAAA and its function.

Ref cougnon et al. is listed 2 times. Please verify all references.

Reviewer 2 Report

The manuscript “Role of eIF5A in mitochondrial function” offers an up-to-date overview of the several connections between the translation factor eIF5A and mitochondria presenting an in-depth description of the different processes in which the protein is involved and contributing to the clarification of the sometimes-contradictory results. The review is well written and organized, comprehensive, and fills a gap in the literature.
